# Carnitine Shuttle and Ferroptosis in Cancer

**DOI:** 10.3390/antiox14080972

**Published:** 2025-08-08

**Authors:** Ye-Ah Kim, Yoonsung Lee, Man S. Kim

**Affiliations:** 1Translational-Transdisciplinary Research Center, Clinical Research Institute, Kyung Hee University Hospital at Gangdong, College of Medicine, Kyung Hee University, Seoul 05278, Republic of Korea; yeak426@khu.ac.kr (Y.-A.K.); ylee3699@khu.ac.kr (Y.L.); 2Department of Biomedical Science and Technology, Graduate School, Kyung Hee University, Seoul 02453, Republic of Korea

**Keywords:** carnitine shuttle, ferroptosis, cancer, lipid peroxidation, CPT1A, redox homeostasis

## Abstract

Ferroptosis is a unique type of regulated cell death characterized by iron-dependent lipid peroxidation, and it has emerged as a promising therapeutic target in cancer treatment. The carnitine shuttle system, which is crucial for transporting fatty acids across the mitochondrial membrane, has been identified as a key regulator of ferroptosis in cancer cells. This review investigates the intricate relationship between the carnitine shuttle and ferroptosis in cancer. We provide a comprehensive review of how the components of the carnitine system, including carnitine palmitoyltransferase 1A (CPT1A), carnitine palmitoyltransferase 2, and carnitine-acylcarnitine translocase, influence cellular redox homeostasis, fatty acid metabolism, and interact with proteins related to ferroptosis sensitivity. We discuss therapeutic implications of targeting the carnitine shuttle system, particularly CPT1A, to overcome ferroptosis resistance and enhance the efficacy of immunotherapy in various cancer types. This review offers further research directions, highlighting the crosstalk between the carnitine shuttle, ferroptosis, and various signaling pathways involved in cancer progression to improve cancer treatment.

## 1. Introduction

Cancer remains a leading cause of mortality, with an estimated 1.96 million new cases and 609,820 deaths projected in the United States in 2023 [1]. Despite advances in conventional therapies such as surgery, chemotherapy, and radiation, many cancers develop resistance to treatment. This underscores the need to identify the molecular pathogenesis and novel therapeutic approaches.

Employing alternative cell death mechanisms has recently emerged as a promising strategy for cancer treatment. Ferroptosis, firstly described by Dixon et al. in 2012, is an iron-dependent regulated cell death, distinct from apoptosis, necroptosis, and autophagy [2]. This unique mode of cell death is driven primarily by iron-dependent lipid peroxidation, which results in oxidative damage to the cell membranes [3]. Ferroptotic cells exhibit morphological alterations, such as condensed mitochondria with decreased cristae and increased membrane density, highlighting the critical role of the mitochondria in this process [4].

Mitochondria are fundamental in energy production and diverse cellular processes, including redox homeostasis, calcium signaling, and cell death regulation [5]. The hyperpolarization of the mitochondrial membrane is emerging as a critical determinant of ferroptosis sensitivity [6].

The carnitine shuttle consists of several key components: carnitine palmitoyltransferase 1 (CPT1), which is localized at the outer mitochondrial membrane; carnitine-acylcarnitine translocase (CACT), which spans the inner mitochondrial membrane (IMM); and carnitine palmitoyltransferase 2 (CPT2), which is situated at the IMM [7]. This shuttle system utilizes L-carnitine as a cofactor to transport activated long-chain fatty acids across the impermeable IMM for subsequent β-oxidation, generating acetyl-CoA for the tricarboxylic acid (TCA) cycle (Figure 1) [8].

Notably, a study by Ma et al. demonstrated that carnitine palmitoyltransferase 1A (CPT1A) acts collaboratively with L-carnitine derived from tumor-associated macrophages (TAMs) to drive ferroptosis resistance in lung cancer stem cells (CSCs) [9]. This discovery has opened new avenues for targeting ferroptosis through modulation of the carnitine shuttle in cancer therapy.

This review explored the intricate relationship between the carnitine shuttle and ferroptosis in cancer. We discuss the molecular mechanisms underlying this relationship, role of L-carnitine derived from the tumor microenvironment, and therapeutic potential of targeting the carnitine shuttle to overcome ferroptosis resistance in cancer treatment.

## 2. Methods

### 2.1. Literature Search Strategy

A comprehensive literature search was conducted in PubMed, Google Scholar, and Web of Science databases to identify relevant studies published from January 2010 to July 2025. However, studies published prior to this period were also included if they were closely related to the content or provided strong foundational evidence. The main search terms, used individually or in combination, included the following: “carnitine shuttle”, “carnitine palmitoyltransferase”, “ferroptosis”, “iron-dependent cell death”, “lipid peroxidation”, “mitochondrial membrane”, “cancer metabolism”, “tumor microenvironment”, and “immunotherapy”. Boolean operators (AND, OR) were used to refine the search results. Following the broad literature search, more detailed studies related to carnitine shuttle and ferroptosis in cancer were sought using keywords such as “Tumor-associated macrophages”, “c-MYC”, “CPT1A”, and “OCTN2”.

### 2.2. Inclusion and Exclusion Criteria

Studies were included if they met one or more of the following conditions:(1)Investigations into the relationship between carnitine shuttle components and ferroptosis in cancer models.(2)Analyses of mitochondrial membrane dynamics in ferroptosis.(3)Explorations of the carnitine shuttle as a therapeutic target in cancer treatment.(4)Discussions on the impact of the tumor microenvironment on carnitine metabolism and ferroptosis.

Studies were excluded if they met one or more of the following criteria:(1)Research exclusively focusing on non-cancer models unrelated to cancer biology.(2)Papers published in languages other than English.(3)Articles lacking sufficient methodological details.(4)Conference abstracts that had not undergone peer review.

### 2.3. Study Quality Assessment

The quality of the included studies was assessed based on the following criteria: validity of experimental design, reproducibility of results, use of appropriate controls, adequacy of statistical analysis, and consistency with existing literature. Priority was given to recently published high-impact studies and those with robust experimental validation in cancer models.

### 2.4. Synthesis of Evidence

The extracted data were used to provide a comprehensive overview of the current state of research and understanding regarding the interplay between the carnitine shuttle and ferroptosis in cancer. Furthermore, information relating to the tumor microenvironment (TME) and its role as a bridge for these interactions is presented. The synthesis primarily focused on highlighting the limitations and future prospects of studies on the connectivity between the carnitine shuttle and ferroptosis.

## 3. Carnitine Shuttle System: Structure and Function

### 3.1. Components and Mechanism of the Carnitine Shuttle

The carnitine shuttle system is crucial for cellular energy metabolism by enabling the transport of long-chain fatty acids into the mitochondrial matrix for β-oxidation to produce energy. Since the IMM is impermeable to fatty acyl-CoA molecules, the shuttle relies on a series of coordinated steps to transport these fatty acids across the membrane.

The initial step of the carnitine shuttle involves the conversion of long-chain fatty acyl-CoA to acylcarnitine by CPT1, located on the outer mitochondrial membrane. There are three tissue-specific types of this enzyme: CPT1A (predominantly expressed in the liver, but also in many other tissues), CPT1B (found mainly in muscle), and CPT1C (expressed primarily in the brain) [10]. CPT1A serves as the rate-limiting enzyme of the carnitine shuttle and regulates the entry of fatty acids into the mitochondria.

Following the formation of acylcarnitine, CACT (also known as SLC25A20) facilitates the transport of acylcarnitine across the IMM via an antiport mechanism. CACT exchanges acylcarnitine in the intermembrane space with free carnitine in the mitochondrial matrix, maintaining a balanced ratio of these compounds on both sides of the membrane [11].

Once inside the mitochondrial matrix, CPT2, which is localized on the inner surface of the IMM, catalyzes the conversion of acylcarnitine back into free carnitine and fatty acyl-CoA [12]. The released fatty acyl-CoA then enters the β-oxidation pathway, where it undergoes sequential oxidation to generate acetyl-CoA, NADH, and FADH2. Free carnitine is transported back to the intermembrane space by the CACT for reuse during the shuttle process.

### 3.2. Regulation of the Carnitine Shuttle

The carnitine shuttle system is tightly regulated to ensure coordinated cellular fatty acid metabolism with energy demand. Principal regulation occurs at the level of CPT1, which is primarily inhibited by malonyl-CoA, the first intermediate in fatty acid synthesis (Figure 2) [13]. This regulatory mechanism ensures the suppression of fatty acid oxidation (FAO) during fatty acid synthesis, thereby preventing a futile cycle.

The intracellular concentration of malonyl-CoA is regulated by the balance between its synthesis by acetyl-CoA carboxylase (ACC), and its degradation by malonyl-CoA decarboxylase (Figure 2) [13]. ACC activity is regulated by various signaling pathways, including the AMP-activated protein kinase pathway, which phosphorylates and inactivates ACC in response to energy depletion [13]. This regulation allows increased FAO during periods of energy shortage.

In addition to allosteric regulation, carnitine shuttle components are subject to transcriptional regulation. Peroxisome proliferator-activated receptors (PPARs), particularly PPARα, are essential in upregulating the expression of CPT1 and other FAO enzymes (Figure 2) [14]. In addition, post-translational modifications, including phosphorylation and acetylation, modulate the activity of carnitine shuttle components.

The availability of L-carnitine, an essential cofactor of the shuttle, represents another regulatory layer. L-carnitine can be synthesized endogenously from lysine and methionine, primarily in the liver and kidney, or obtained from dietary sources [15]. The cellular uptake of L-carnitine is mediated by specific transporters such as organic cation transporter novel 2 (OCTN2), which is responsible for high-affinity carnitine transport across the plasma membrane [16].

### 3.3. Carnitine Shuttle in Cancer Metabolism

Cancer cells exhibit remarkable metabolic plasticity, enabling the adaptation of their energy production pathways based on nutrient availability and environmental conditions. The Warburg effect, which is characterized by increased glucose uptake and lactate production, even in the presence of oxygen, has traditionally been considered a hallmark of cancer metabolism, and accumulating evidence indicates that many cancer cells rely on FAO for energy production, particularly under nutrient-limited conditions [17]. Recent studies have focused on CD36 as a target for cancer as it is involved in fatty acid transport and metabolism during cancer metastasis [18,19].

The carnitine shuttle system plays a crucial role in cancer cell growth and survival by facilitating FAO. Multiple studies have reported the altered expression of carnitine shuttle components in various cancer types. For instance, overexpression of CPT1A has been observed in prostate cancer, colorectal cancer, breast cancer, and glioblastoma [20,21,22,23,24]. This upregulation is associated with enhanced FAO, providing cancer cells with an alternative energy source [25].

In addition, the carnitine shuttle system influences various aspects of cancer progression. CPT1A has been implicated in tumor initiation, metastasis, and therapy resistance [21,26,27]. In breast cancer cells, CPT1 promotes cell survival under metabolic stress by maintaining redox homeostasis and ATP production related to electron transport chain (ETC) [28]. In colorectal cancer, CPT1A enhances metastatic potential by supporting cancer stem cell properties [21]. Additionally, CPT1A overexpression has been associated with chemoresistance in multiple cancer types, and it has been observed that cancer cells surviving radiotherapy and chemotherapy enhance fatty acid oxidation through CPT1 and CPT2 [29,30]. This suggests that aggressive, highly metastatic, and therapy-resistant cancer cells exhibit a strong reliance on FAO, and targeting this vulnerability is akin to cutting off the emergency fuel supply to inhibit growth due to energy depletion.

Considering its pivotal role in cancer metabolism, the carnitine shuttle system has emerged as a promising therapeutic target. CPT1 inhibitors such as etomoxir and perhexiline have shown promising anti-cancer effects in preclinical studies [31]. However, their clinical application is limited by off-target effects and potential toxicity. More selective approaches targeting specific isoforms or downstream effectors of the carnitine shuttle pathway are currently under investigation, aiming to disrupt cancer metabolism while minimizing the adverse effects on normal cells.

## 4. Ferroptosis: Mechanisms and Regulation

### 4.1. Molecular Mechanisms of Ferroptosis

Ferroptosis is a form of regulated cell death characterized by iron-dependent accumulation of lipid peroxides, leading to oxidative damage to cellular membranes [6]. Unlike other forms of cell death, such as apoptosis or necroptosis, ferroptosis does not involve caspase activation, mitochondrial cytochrome c release, or plasma membrane rupture [6]. Instead, it exhibits unique morphological and biochemical features, including mitochondrial shrinkage, increased membrane density, reduced cristae, and accumulation of lipid reactive oxygen species (ROS) [6].

Ferroptosis typically involves three interconnected pathways: iron homeostasis, redox homeostasis, and lipid metabolism. Iron plays a crucial role in generating lipid ROS through Fenton reactions, where ferrous iron (Fe^2+^) reacts with hydrogen peroxide to produce highly reactive hydroxyl radicals [32]. These radicals can attack polyunsaturated fatty acids (PUFAs) in membrane phospholipids, triggering lipid peroxidation chain reaction (Figure 1) [33].

Regarding lipid metabolism, another key determinant of ferroptosis sensitivity is the cellular content of PUFA-containing phospholipids, which are primary targets of lipid peroxidation [34]. The biosynthesis of PUFA-phospholipids involves several enzymes, including acyl-CoA synthetase long-chain family member 4 (ACSL4) and lysophosphatidylcholine acyltransferase 3 (LPCAT3) [35]. ACSL4 activates PUFAs by forming PUFA-CoA, whereas LPCAT3 incorporates them into membrane phospholipids, particularly phosphatidylethanolamines [36].

The glutathione peroxidase 4 (GPX4) system is the primary defense mechanism related to redox homeostasis against ferroptosis [37]. GPX4, a selenoenzyme, facilitates the reduction of lipid hydroperoxides into nontoxic lipid alcohols using reduced glutathione (GSH) as a cofactor [38]. The depletion of GSH or direct inhibition of GPX4 increases cellular susceptibility to ferroptosis by impairing their ability to neutralize lipid peroxides (Figure 1) [39].

Recent studies have identified additional ferroptosis regulatory mechanisms, including the ferroptosis suppressor protein 1 (FSP1)-coenzyme Q10 (CoQ10) axis [40,41] and GPX4-independent pathways [42]. FSP1 is located in the plasma membrane and reduces CoQ10 to ubiquinol, which acts as a lipophilic radical trapping antioxidant and provides a ferroptosis defense system [40]. Additionally, the GTP cyclohydrolase 1-tetrahydrobiopterin (BH4) pathway has been identified as a third ferroptosis defense system, wherein BH4 acts as a radical trapping antioxidant independent of GSH or CoQ10 [43].

### 4.2. Role of Mitochondria in Ferroptosis

Although the clear role of mitochondria in ferroptosis remains uncertain, accumulating evidence suggests their involvement in various aspects of ferroptosis. Morphological changes of mitochondria, distinct from those observed in other cell death types, highlight the unique impact of ferroptosis on mitochondrial structure, implying interplay between ferroptosis and mitochondria.

Mitochondria serve as significant sources of ROS production and iron utilization, both of which are critical factors in ferroptosis initiation [44]. The ETC in mitochondria generates superoxide radicals, which are subsequently converted into hydrogen peroxide [45]. In the presence of iron, hydrogen peroxide can form hydroxyl radicals through Fenton reactions, thereby amplifying oxidative damage [46]. Additionally, mitochondria contain numerous iron-containing proteins such as iron–sulfur-cluster-containing enzymes and cytochromes, which contribute to the cellular iron pool [47].

Mitochondrial membrane components, particularly cardiolipin and other phospholipids, influence ferroptosis sensitivity [6]. Cardiolipin—a unique PUFA-rich phospholipid localized to the IMM—is highly prone to peroxidation [48]. Oxidized cardiolipin impairs mitochondrial function, leading to ROS production and oxidative damage, and amplifies the ferroptotic signal [49].

Recent studies have highlighted the role of mitochondrial fatty acid metabolism in modulating ferroptosis [50]. The mitochondrial fatty acid synthesis pathway has been implicated in ferroptosis regulation, and disruption of this pathway enhances ferroptosis susceptibility [2]. Conversely, the carnitine shuttle system, which facilitates the transport of fatty acids into mitochondria for β-oxidation, influences ferroptosis in a context-dependent manner [51].

Furthermore, mitochondrial dynamics, including fusion, fission, and mitophagy, also affect ferroptosis sensitivity, which is related to nuclear factor erythroid 2-related factor 2 (NRF2) and mitochondrial fission mediator dynamin-related protein 1 (DRP1) [52]. Mitochondrial fragmentation, associated with increased fission, has been observed during ferroptosis [53,54]. Additionally, mitophagy, the selective degradation of damaged mitochondria, can mitigate ferroptosis by eliminating ROS and oxidative damage [55].

### 4.3. Ferroptosis in Cancer

Ferroptosis has emerged as a promising strategy in cancer therapy, particularly for malignancies resistant to conventional treatments [56]. Cancer cells often exhibit alterations of iron metabolism, redox homeostasis, and lipid composition, affecting their ferroptosis susceptibility [57,58]. For example, increased iron uptake and storage are observed in several cancer types, potentially causing cancer cells to be more vulnerable to iron-dependent oxidative damage, and ferroptosis [59,60,61].

Multiple therapeutic approaches have been explored to induce ferroptosis in cancer cells [62]. One of the approaches is direct inhibition of GPX4 using small molecules such as RSL3 [63]. However, clinical applications of the compounds for inhibition of GPX4 are limited due to their narrow therapeutic windows and potential toxicity [64]. Alternative approaches include GSH depletion using compounds such as erastin or indirectly inhibiting GPX4 through system Xc- inhibition [65].

The enhancement of ferroptosis in combination with established therapies has shown promising results in preclinical studies [66]. Radiotherapy, which increases cellular ROS and iron levels, acts synergistically with ferroptosis inducers to induce cancer cell death [62]. Similarly, immunotherapy, particularly with immune checkpoint inhibitors, has been shown to promote tumor ferroptosis by enhancing T cell-mediated lipid peroxidation [67].

The ferroptosis sensitivity of CSCs has been a research focus [68]. CSCs, responsible for tumor initiation, progression, and therapy resistance, often exhibit distinct metabolic profiles compared to those of bulk tumor cells [69]. Recent studies have revealed that some CSCs are more resistant to ferroptosis due to antioxidant system enhancement or metabolic reprogramming, suggesting that targeting these resistance mechanisms improves cancer therapy [70].

## 5. Interconnection Between Carnitine Shuttle and Ferroptosis in Cancer

### 5.1. Mitochondrial Membrane Dynamics in Ferroptosis

The mitochondrial membrane is essential in ferroptosis and serves as a primary site for lipid peroxidation and regulator of cellular redox homeostasis [6]. Mitochondrial membranes, particularly the IMM, are rich in PUFAs, making them susceptible to lipid peroxidation during ferroptosis [71,72]. This leads to structural and functional alterations, including decreased membrane potential, compromised respiratory chain activity, and ultimately cell death [73,74].

The lipid composition of mitochondrial membranes can modulate ferroptosis sensitivity. Cardiolipin, a unique phospholipid predominantly found in the IMM, contains a high proportion of PUFAs and is crucial in mitochondrial structure and function [75]. Cardiolipin oxidation during ferroptosis disrupts mitochondrial cristae organization and impairs ETC complexes, leading to increased ROS production and oxidative damage [74,75,76,77].

Under normal conditions, mitochondria maintain a dynamic equilibrium between fusion and fission, regulated by specific proteins, such as mitofusins, optic atrophy 1, and DRP1 (Figure 3) [78]. During ferroptosis, increased mitochondrial fragmentation is observed, suggesting a shift toward fission [79]. This fragmentation may facilitate lipid peroxidation signals and mitochondrial dysfunction [6].

The mitochondrial permeability transition pore (mPTP), a nonselective channel spanning the inner and outer mitochondrial membranes, contributes to ferroptosis [80]. Opening the mPTP can lead to mitochondrial swelling, rupture of the outer membrane, and release of pro-death factors. Recent studies have suggested that mPTP opening occurs during ferroptosis, potentially as a result of lipid peroxidation-induced membrane damage or calcium dysregulation [80].

### 5.2. Carnitine Shuttle Influence on Lipid Peroxidation and Ferroptosis

The carnitine shuttle system exerts multifaceted effects on lipid peroxidation and ferroptosis via complex interactions with multiple cellular defense mechanisms. By facilitating the transport of long-chain fatty acids into mitochondria for FAO, the carnitine shuttle impacts the availability of substrates for lipid peroxidation and generation of reducing equivalents that maintain cellular antioxidant systems [81].

The presence of membrane transporters mediating the intracellular and extracellular transport of carnitine, OCTN2, enables intercellular carnitine exchange, suggesting a molecular basis for cancer cells to absorb carnitine from the microenvironment (Figure 3). Indeed, overexpression of OCTN2 has been observed in glioblastoma [82], breast cancer [83], and ovarian cancer [84].

A rate-limiting enzyme in the carnitine shuttle, CPT1A, has emerged as a key regulator of ferroptosis sensitivity in multiple cancer types [21], which is attributed to several mechanisms, including the decreased production of NADPH (a critical cofactor for glutathione regeneration), altered lipid composition, and increased mitochondrial ROS production [25]. The upregulation of CPT1A, which is commonly observed in aggressive cancers, protects against ferroptosis by promoting FAO [21]. This protective effect is mediated by the enhanced generation of reducing equivalents (NADPH, NADH, and FADH2) during FAO, which can be utilized for antioxidant defense [85,86]. Additionally, increased FAO reduces the cellular polyunsaturated fatty acids, thereby shrinking the availability of lipid peroxidation substrates [87].

The effect of the carnitine shuttle on ferroptosis extends beyond CPT1A to other system components. CACT in the IMM has been implicated in the maintenance of mitochondrial function and prevention of oxidative damage [88]. Similarly, CPT2 contributes to the overall efficiency of FAO and energy production, indirectly affecting cellular resilience to oxidative stress [23].

L-carnitine, an essential cofactor of the carnitine shuttle, exhibits antioxidant properties that may protect against ferroptosis [9]. As a small molecule with a quaternary ammonium group, L-carnitine scavenges free radicals and disturbs lipid peroxidation [81]. Additionally, L-carnitine supplementation enhances mitochondrial function and reduces oxidative stress [89]. However, the role of L-carnitine in cancer-related ferroptosis is complex and context-dependent.

A positive feedback loop exists between CPT1A and the oncogenic transcription factor c-Myc, which is crucial for maintaining cancer stem cell properties and preventing ferroptosis [9]. This loop operates through a dual mechanism where CPT1A stabilizes c-Myc, and stabilized c-Myc, in turn, promotes CPT1A expression [9]. This self-reinforcing loop enhances the cell’s antioxidant defense system (e.g., increased GPX4 activity) and reduces substances susceptible to lipid peroxidation (e.g., decreased PUFA content through ACSL4 downregulation), thereby increasing resistance to ferroptosis [9]. The importance of GPX4 in ferroptosis resistance has been extensively documented in several cancer types [34,53]. These regulatory mechanisms are supported by studies showing that ACSL4 is a key determinant of ferroptosis sensitivity in various cancer cell lines [35]. Targeting the CPT1A/c-Myc loop is considered a promising therapeutic strategy for overcoming ferroptosis resistance in cancer cells and enhancing the efficacy of immunotherapies. However, research elucidating the relationship between CPT1A and ferroptosis is still limited, and its relationship has been found in a certain cancer type, lung cancer. More studies are needed to strongly establish its connection with the carnitine shuttle, and other types of cancer.

### 5.3. Interconnection with Tumor-Associated Macrophages

The interaction between tumor-associated macrophages (TAMs) and cancer cells within the tumor microenvironment is crucial for tumorigenesis and is also known to play a key role in tumor metastasis. For instance, in the process of colorectal cancer metastasis, it has been revealed that TAMs promote this process through the regulation of the JAK2/STAT3/miR-506-3p/FoxQ1 axis [90]. Furthermore, c-Myc is also expressed in TAMs, and inhibiting c-MYC activity has been confirmed to block the expression of pro-tumorigenic genes [91].

The research by Ma et al. has revealed the crucial role of the tumor microenvironment in modulating ferroptosis sensitivity through the carnitine shuttle system [9]. It is particularly noteworthy that TAMs provide a significant source of L-carnitine to cancer cells, contributing to their ferroptosis resistance (Figure 3) [9].

TAMs with an M2-like phenotype express high levels of γ-butyrobetaine hydroxylase (BBOX1), a key enzyme in L-carnitine biosynthesis [9,92]. These macrophages synthesize and secrete L-carnitine into the tumor microenvironment, which is taken up by cancer cells, particularly CSCs [9]. This uptake process is facilitated by OCTN2, which supports CSCs enhancement [9,93]. Differentiation of macrophage induces OCTN2-mediated L-carnitine transport through stimulation of the mTOR-STAT3 axis, suggesting that the polarization of macrophages directly affects their carnitine-handling capacity [94].

However, the interaction between macrophages and cancer cells is not unidirectional. Mechanisms by which macrophages differentiate into TAMs through ferroptosis in cancer cells have also been introduced [95]. Furthermore, a recent study showed that enhancing ferroptosis in TAMs can induce potent anti-tumor activity, which has the effect of inducing mitochondrial changes in TAMs, converting them into an activated state with tumor-killing capabilities [96].

### 5.4. Interactions of Signaling Pathways with Carnitine Shuttle and Ferroptosis

The Wnt/β-catenin signaling pathway plays a crucial role in suppressing ferroptosis through the transcriptional regulation of GPX4 [97]. Although studies on the direct association with the carnitine shuttle system are limited, a potential link is suggested between cellular metabolic programs regulated by Wnt/β-catenin and mitochondrial metabolic homeostasis maintained by CPT1A [98]. Furthermore, the Hippo/YAP signaling pathway regulates ferroptosis in a cell-density-dependent manner, and YAP upregulates ferroptosis regulators (such as ACSL4, TFRC), influencing lipid reprogramming [99,100]. These pathways are also expected to contribute to the metabolic plasticity and therapeutic resistance of cancer cells through indirect regulation of fatty acid metabolic pathways, including the carnitine shuttle.

The PI3K/Akt/mTOR signaling pathway plays a pivotal role in regulating the carnitine shuttle system and ferroptosis, and the complex interactions among them critically influence cancer cell metabolic adaptation and survival. Firstly, PI3K/Akt signaling directly regulates fatty acid oxidation by inhibiting CPT1A (carnitine palmitoyltransferase 1A) expression [101]. Interestingly, under growth-factor-rich conditions, PI3K signaling is required to inhibit lipid catabolism, which is achieved through CPT1A expression suppression [101]. This regulatory mechanism is a primary way for cells to inhibit beta-oxidation during anabolic growth; if CPT1A is constitutively expressed, growth factor stimulation fails to suppress beta-oxidation [101]. Meanwhile, in relation to ferroptosis, the oncogenic activation of the PI3K/Akt/mTOR pathway suppresses ferroptosis through SREBP1 (sterol regulatory element-binding protein 1)-mediated lipogenesis [102]. Specifically, sustained activation of mTORC1 and mTORC1-dependent SREBP1 induction are required, and SCD1 (stearoyl-CoA desaturase-1), a transcriptional target of SREBP1, produces monounsaturated fatty acids, mediating SREBP1’s ferroptosis-inhibiting activity [102]. This mechanism acts in conjunction with another protective mechanism where mTORC1 promotes GPX4 protein synthesis, preventing lipid peroxidation and acting as a “protective shield” against ferroptosis [103].

## 6. Therapeutic Implications

### 6.1. Targeting Carnitine Shuttle to Induce Ferroptosis in Cancer

The link between carnitine shuttle and ferroptosis presents a potential target for therapeutic intervention in cancer [20]. A comprehensive review of the current therapeutic agents targeting the carnitine shuttle system revealed diverse approaches with varying degrees of clinical development and efficacy profiles (Table 1).

Among direct CPT1A inhibitors, etomoxir has been extensively studied in multiple cancer types, including prostate [23], breast [28], leukemia [31], and bladder cancers [87]. This irreversible inhibitor formed covalent adducts with CPT1A, effectively blocking FAO and enhancing sensitivity to ferroptosis (Table 1). However, clinical translation remains limited by significant hepatotoxicity and off-target effects on the mitochondrial complex I [104]. Thus, the narrow therapeutic window of etomoxirs underscores the need for the development of safer alternatives.

Perhexiline is a promising second generation CPT1A inhibitor with improved safety profiles compared to etomoxir (Table 1). As a reversible inhibitor that targets both CPT1A and CPT1B, perhexiline maintains anti-cancer efficacy while demonstrating reduced cardiotoxicity in preclinical chronic lymphocytic leukemia (CLL) and hepatocellular carcinoma (HCC) [105,106]. However, its limitations and the requirement for therapeutic drug monitoring present challenges for its clinical implementation.

The discovery of TAM-derived L-carnitine as a critical ferroptosis resistance mechanism has opened new therapeutic approaches. A competitive inhibitor of BBOX1, mildronate (meldonium), effectively reduced biosynthesis of L-carnitine in TAMs. Clinical studies on lung cancer [9] and hepatocellular carcinoma [107] demonstrated that mildronate decreases carnitine availability and enhances immunotherapy responses. Notably, the established safety profile of mildronate as a cardiac medication provides a significant advantage for rapid clinical translation, although combination therapeutic approaches are typically required to achieve optimal efficacy.

ST1326 represents a breakthrough as a brain-penetrant-selective CPT1A inhibitor designed for glioblastoma treatment (Table 1). This agent demonstrated selectivity for cancer cells over normal brain tissue, addressing a critical unmet need in neuro-oncology [110]. However, the early developmental stage limits the available efficacy data.

Identification of the CPT1A/c-Myc positive feedback loop has led to novel therapeutic approaches targeting this regulatory circuit [9]. Additionally, a positive feedback loop between EBP2 and c-Myc has been revealed, implying that targeting c-Myc can be a promising clinical strategy with benefits [116]. Since it has been considered not to be applicable for tumor treatment, it has been newly conducted in a phase 1 study, named ‘OMO-103’ recently [111]. While complex protein delivery requirements and potential immunogenicity concerned challenges to the medication, a recent study revealed preliminary evidence of clinical benefit of OMO-103 [111].

Lastly, etoposide is a topoisomerase II inhibitor, and works secondly towards the non-competitive inhibition of OCTN2-mediated carnitine transport [115]. Despite its primary mechanism, DNA damage, it disrupts carnitine shuttle at higher concentrations [104]. Its clinical application has been conducted on lung cancer [112], lymphomas [113], and leukemias [114].

However, challenges still remain for clinical translation, linking ferroptosis and carnitine shuttle system to treat cancers. Therapeutic efficacy and safety are needed to be implicated.

### 6.2. Combination with Immunotherapy

The relationship between ferroptosis and antitumor immunity has collected significant attention in recent years, revealing promising opportunities for combination therapies [67]. Emerging evidence suggests that ferroptosis enhances the efficacy of immunotherapy, particularly immune checkpoint inhibitors, through multiple mechanisms [9].

A groundbreaking study by Ma et al. demonstrated that targeting CPT1A sensitized lung cancer to immune checkpoint blockades [9]. Mechanistically, CPT1A inhibition enhances tumor ferroptosis, promoting the infiltration and activation of CD8+ T cells within the tumor microenvironment [9]. This increased T-cell activity further amplifies ferroptosis, creating a positive feedback loop that potentiates antitumor immune responses [9].

In addition, targeting the carnitine shuttle can modulate other aspects of the tumor immune microenvironment [67]. CPT1A inhibition reduces the proportion of immunosuppressive cell populations, for example, M2-like TAMs and myeloid-derived suppressor cells [9]. Additionally, disruption of the L-carnitine supply chain by targeting BBOX1 in TAMs repolarizes these cells toward a more pro-inflammatory and anti-tumor phenotype [117].

The immunogenicity of ferroptotic cell death also contributes to the synergy between carnitine shuttle targeting and immunotherapy [67]. Unlike apoptosis, which is generally considered immunologically silent, ferroptosis is highly immunogenic, releasing various damage-associated molecular patterns that activate dendritic cells and stimulate adaptive immune responses [118]. This immunogenic effect may enhance the expansion of tumor-specific T cells, thereby potentiating the efficacy of checkpoint inhibitors [9].

Clinical translation of the immunotherapy combination strategies is in progress, with several trials evaluating the safety and efficacy of ferroptosis inducers in combination with immunotherapies [119]. Although specific trials targeting the carnitine shuttle in combination with immunotherapy are still limited, the strong preclinical rationale and preliminary clinical data from related approaches suggest significant potential for this strategy [9].

Future developments in this field may include more selective inhibitors of carnitine shuttle components, particularly those that can differentially target cancer cells and immune cells [2]. Additionally, biomarker-guided patient selection based on the expression of CPT1A, c-Myc, or other pathway components may help identify those most likely to benefit from these combination approaches [9].

### 6.3. Clinical Development and Challenges

However, clinical translation faces several challenges. Given the reported significant hepatotoxicity associated with early compounds such as etomoxir, the development of selective inhibitors with improved safety profiles is critically important [120]. Isoform-specific inhibitors that preferentially target CPT1A over CPT1B or CPT1C are essential to minimize potential cardiac and skeletal muscle toxicity [120].

Patient selection is also a critical consideration in clinical development. Considering the heterogeneity of cancer metabolism, tumors do not depend equally on the carnitine shuttle for survival and ferroptosis resistance [7]. Identifying biomarkers that predict sensitivity to carnitine shuttle inhibition, such as the high expression of CPT1A, c-Myc, and OCTN2, or specific metabolic signatures, could help stratify patients and optimize therapeutic outcomes [9]. Integrating multi-omics approaches, including genomics, transcriptomics, and metabolomics, may provide comprehensive biomarker profiles for precision medicine applications [121]. Monitoring treatment responses presents a challenge, as ferroptosis may not immediately manifest as tumor shrinkage. The development of reliable pharmacodynamic indicators, such as lipid peroxidation products or specific imaging tracers, should facilitate the evaluation of treatment efficacy [122].

Concerns regarding systemic toxicity, including cardiomyopathy, muscle weakness, and metabolic disturbances, must be addressed using strategies such as intermittent dosing, targeted drug delivery, and organ function monitoring [120,123,124]. Furthermore, understanding adaptive resistance mechanisms, such as the activation of alternative pathways and enhanced antioxidant systems, will be crucial for developing effective therapeutic strategies [9,21].

Resistance to ferroptosis-inducing therapeutic agents represents a significant challenge in cancer treatment. This resistance can primarily arise through the activation of compensatory antioxidant systems and metabolic rewiring within cancer cells. For instance, the SLC7A11/GPX4 pathway is central to ferroptosis resistance, and increased GPX4 expression can effectively suppress lipid peroxidation, thereby inducing resistance [39]. Additionally, cancer cells reduce ferroptosis sensitivity by altering iron, lipid, glucose, and glutamine metabolism [125]. These metabolic alterations, coupled with the upregulation of anti-ferroptotic proteins such as ferritin, form a robust defense mechanism against ferroptosis inducers [58]. Overcoming these complex resistance mechanisms is essential for the development of ferroptosis-based cancer therapeutics.

## 7. Future Research Directions

The intersection of the carnitine shuttle and ferroptosis in cancer is a rapidly evolving field, pointing to several promising research directions [3]. As our understanding of these processes deepens, further exploration is needed to advance both fundamental scientific knowledge and clinical applications.

First, we still need to fully understand the precise mechanisms by which the carnitine shuttle influences mitochondrial membrane dynamics and lipid peroxidation [126]. Although a general relationship between fatty acid metabolism and ferroptosis sensitivity has been established, we need more research on the specific lipid species and membrane domains affected by carnitine shuttle activity [6]. Advanced lipidomic approaches coupled with high-resolution microscopy techniques may reveal the spatial and temporal dynamics of lipid peroxidation related to the mitochondrial membrane composition [35].

Second, it is crucial to consider the broader metabolic network connected to the carnitine shuttle [7]. The carnitine shuttle operates within a complex metabolic environment, interacting with various pathways, including glycolysis, the TCA cycle, and amino acid metabolism [8]. Understanding how these pathways mutually affect ferroptosis sensitivity may reveal additional therapeutic targets and strategies [20]. Metabolic flux analysis can offer a powerful approach for mapping these interconnections [126].

Third, the relationship between ferroptosis, the carnitine shuttle, and CSCs is particularly noteworthy [9]. CSCs exhibit a distinct ferroptosis sensitivity compared to regular tumor cells, partly because of altered carnitine shuttle activity [68]. Identifying the metabolic and ferroptotic processes in CSCs across different cancer types may reveal promise for therapeutic exploitation [8].

Fourth, we should also closely examine the crosstalk among the carnitine shuttle, ferroptosis, and signaling pathways involved in cancer progression [21]. Recent studies have highlighted connections between metabolic pathways and known oncogenic signaling networks, such as PI3K/Akt/mTOR, Wnt/β-catenin, and Hippo/YAP [126]. Exploring how these signaling pathways influence or are influenced by the carnitine shuttle and ferroptosis, and how they regulate these processes, is expected to reveal novel regulatory mechanisms that could lead to future cancer treatment strategies [127].

Finally, integrating genomic, transcriptomic, proteomic, and metabolomic profiles with clinical outcomes may help develop predictive models for patient classification and treatment selection [127]. These analytical approaches can accelerate the clinical translation of therapies that target the carnitine shuttle, and induce ferroptosis, helping to identify optimal patient populations and combination strategies [127].

## 8. Conclusions

This review explores the molecular mechanisms underlying the interplay between the carnitine shuttle and ferroptosis, focusing on mitochondrial membrane dynamics and the interactions between cancer cells and their microenvironment, including CSCs and TAMs. We also discuss their implications for cancer biology and therapeutic applications.

The carnitine shuttle has emerged as a critical regulator of ferroptosis, in addition to its role in redox homeostasis within cancer cells. It orchestrates cellular control over lipid peroxidation and ferroptotic cell death by influencing energy metabolism, mitochondrial membrane composition, fatty acid metabolism, and antioxidant systems. Notably, the recently discovered CPT1A/c-Myc positive feedback loop is a significant mechanism that coordinates a comprehensive ferroptosis defense program within CSCs.

The tumor microenvironment, particularly TAMs, plays a critical role in regulating ferroptosis by supplying L-carnitine to cancer cells. This metabolic symbiosis creates an environment conducive to cancer cell survival and treatment resistance. Strategies that target L-carnitine synthesis in macrophages and its uptake and utilization by cancer cells—thereby disrupting their interaction with TAMs—could evolve into future therapies to overcome cancer cell energy supply and ferroptosis resistance.

Targeting the carnitine shuttle to induce ferroptosis offers new therapeutic opportunities for cancer treatment, particularly for malignancies resistant to conventional therapies. Inhibitors of carnitine shuttle components such as CPT1A can sensitize cancer cells to ferroptosis inducers and enhance the efficacy of immunotherapy. This approach may be particularly effective against CSCs, which often rely on an altered metabolism for survival and resistance to therapy.

However, translating this knowledge into clinical practice still faces several challenges. Here are the key hurdles we need to overcome: developing highly selective inhibitors with excellent pharmacokinetic and safety profiles, discovering predictive biomarkers for patient selection, effectively monitoring treatment response, and managing potential systemic toxicities.

Addressing these challenges demands interdisciplinary collaboration and innovative approaches, including advanced drug delivery systems, precise imaging techniques, and computational modeling. We anticipate groundbreaking advancements, especially through integrating new technologies like single-cell omics, high-resolution imaging, gene manipulation techniques, and physiologically relevant preclinical models. This combined approach will deepen our understanding of the carnitine shuttle–ferroptosis axis and accelerate the development of effective therapeutic strategies.

In conclusion, the interconnection between the carnitine shuttle and ferroptosis offers new insights into cancer cell mechanisms and provides novel therapeutic targets. By unraveling the complex relationship, we can establish new paradigms for cancer therapy that target the metabolic vulnerabilities of cancer cells while protecting normal tissues, ultimately contributing to improved prognoses and outcomes in patients with cancer.

## Figures and Tables

**Figure 1 antioxidants-14-00972-f001:**
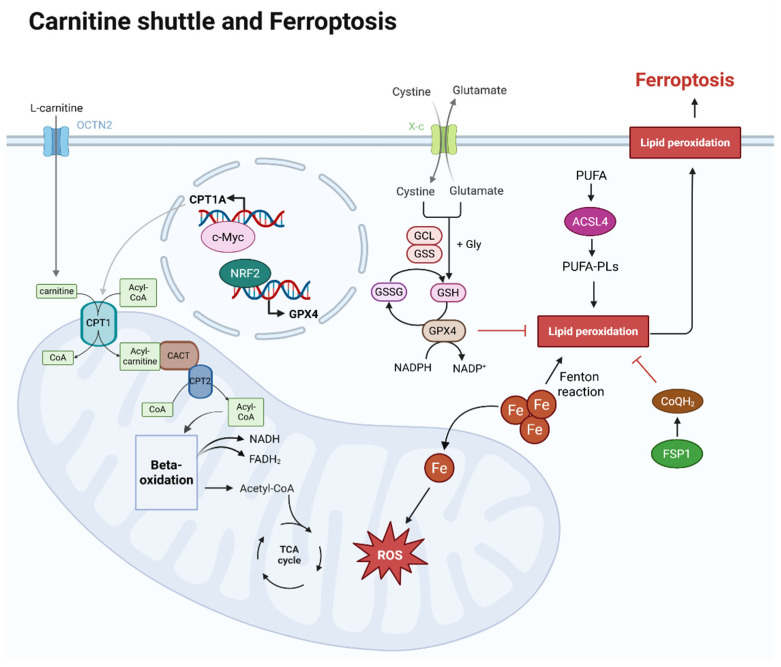
Overview of carnitine shuttle and ferroptosis within a cell. L-carnitine is transported by OCTN2 in the cell membrane, and the carnitine is transported into mitochondria and helps to transport CoA via the carnitine shuttle. Acyl-CoA, linked to long-chain fatty acids by the carnitine shuttle, enters the mitochondrial matrix, where long-chain fatty acids are used as an energy source in beta-oxidation. The acetyl-CoA produced at this stage then enters the TCA cycle, also generating NADH and FADH_2_. At this point, CPT1A expression is regulated by c-Myc. The mechanism of ferroptosis is iron-dependent. As iron flows into the mitochondria, it generates reactive oxygen species (ROS), which can lead to lipid peroxidation. Lipid peroxidation susceptibility increases as polyunsaturated fatty acids (PUFAs) are oxidized and converted into phospholipid (PUFA-PLs) forms. This process is promoted by ACSL4. However, GPX4 and CoQH_2_ can inhibit lipid peroxidation. After cystine uptake and conversion to cysteine, GSH is produced, and both GSH and GPX4 inhibit lipid peroxidation. The expression of GPX4 is regulated by NRF2. CoQH_2_, converted from CoQ by FSP1, also inhibits lipid peroxidation. Created in BioRender. Mol, C. (2025) https://BioRender.com/iv1c7gu.

**Figure 2 antioxidants-14-00972-f002:**
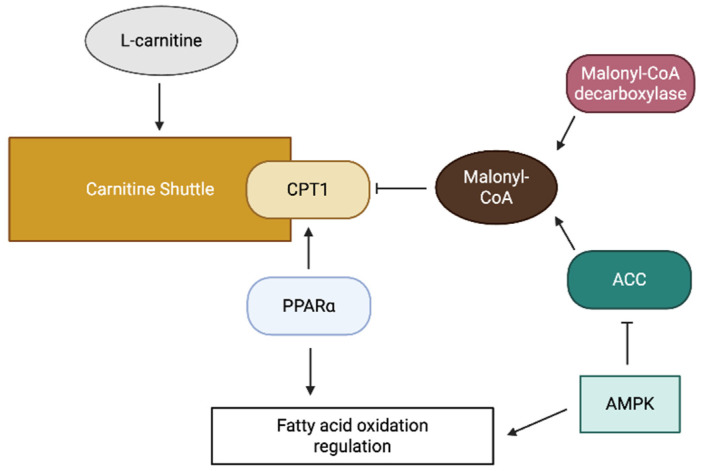
A schematic diagram of carnitine shuttle regulation.

**Figure 3 antioxidants-14-00972-f003:**
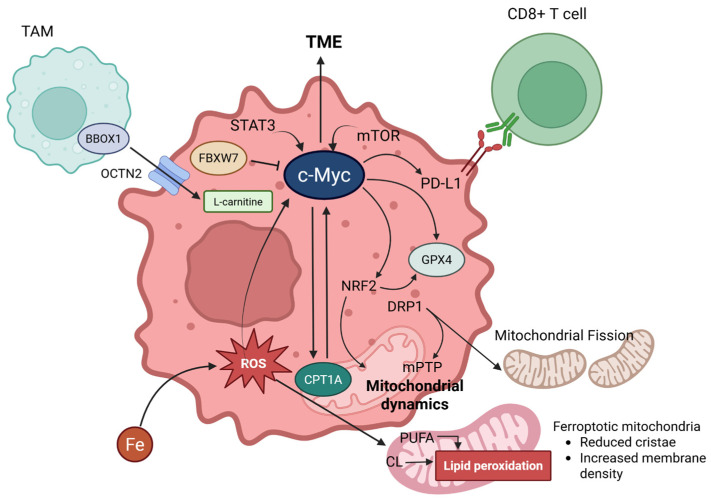
Comprehensive figures for interconnection between carnitine shuttle (CPT1A) and ferroptosis via c-Myc. L-carnitine enters cancer cells via OCTN2, facilitated by BBOX1 in TAMs (Tumor-Associated Macrophages). c-MYC, expressed in cancer cells, is induced by the influence of STAT3 and mTOR signaling pathways, significantly impacting the TME (Tumor Microenvironment). c-MYC engages in a positive feedback loop with CPT1A, promotes the expression of PD-L1 (which binds to CD8+ T cells), and upregulates NRF2 and GPX4. GPX4 is further promoted by NRF2. Additionally, NRF2 is involved in mitochondrial dynamics. DRP1, activated by ferroptosis, can induce mPTP (mitochondrial permeability transition pore) and mitochondrial fission. Lastly, FBXW7 can inhibit c-MYC. Created in BioRender. Mol, C. (2025) https://BioRender.com/hsuyumk.

**Table 1 antioxidants-14-00972-t001:** Clinical therapeutic agents targeting the carnitine shuttle system for cancer treatment.

Agent	Target	Mechanism	Cancer Types	Key Findings	Limitations	References
Etomoxir	CPT1A	Irreversible inhibitor; forms covalent adduct with CPT1A	Prostate [23], breast [28], leukemia [31], bladder [87]	Enhanced ferroptosis sensitivity; reduced proliferation; synergy with chemotherapy	Hepatotoxicity; off-target effects on complex I [104]; narrow therapeutic window	[23,28,31,87,104]
Perhexiline	CPT1A/CPT1B	Reversible inhibitor; better safety profile	CLL [105], Hepatocellular cancer [106]	Safer than etomoxir; maintained anti-cancer efficacy; reduced cardiotoxicity	Limited bioavailability; requires therapeutic drug monitoring	[105,106]
Mildronate (Meldonium)	BBOX1	Competitive inhibitor of L-carnitine biosynthesis	Lung cancer [9], hepatocellular carcinoma [107]	Reduces TAM-derived L-carnitine; enhances immunotherapy response; well-tolerated	Requires combination therapy for efficacy; established cardiac drug	[9,107,108]
ST1326	CPT1A	CPT1A inhibitor, induction of apoptosis	Acute myeloid leukemia [109], CLL [110]	Synergistic effects with ABT199 in leukemia; prevents c-myc-driven tumorigenesis	Non-selective tissue distribution, limited solid tumor data	[109,110]
Omomyc	c-Myc	Dominant-negative c-Myc mutant	Various advanced cancers including non-small cell lung cancer	Disrupts CPT1A/c-Myc loop; shows clinical activity	Complex protein delivery; immunogenicity concerns	[9,111]
Etoposide	OCTN2 (secondary)	Topoisomerase II inhibitor with OCTN2 inhibitory activity	Lung cancer [112], lymphoma [113], leukemia [114]	Inhibits OCTN2-mediated carnitine transport; established anti-cancer agent	Primary mechanism is DNA damage; carnitine effects are secondary	[112,113,114,115]

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
