# Peer review of "Carnitine Shuttle and Ferroptosis in Cancer"

_antioxidants, 2025, doi:10.3390/antiox14080972_

Round 1
Reviewer 1 Report
- While the findings on the CPT1A/c-Myc feedback loop and TAM-derived L-carnitine are significant, the review's discussion in sections 4.3 and 4.4 is almost entirely dependent on a single study by Ma et al. This narrows the scope, making these sections a summary of one paper rather than a comprehensive review of the field. Integrating other related research is necessary to provide a broader and more critical perspective.
- The manuscript would be strengthened by expanding on its observation that the carnitine shuttle's role is "context-dependent." A deeper discussion is needed to explain the central paradox: why is CPT1A upregulation a survival mechanism, yet its inhibition is a viable therapeutic strategy? Explaining how a tumor's specific metabolic phenotype—such as its reliance on fatty acid oxidation (FAO) over glycolysis—creates this targetable vulnerability would offer valuable insight.
- The review's discussion of signaling pathways is narrowly focused on the c-Myc connection. While the "Future Research Directions" section acknowledges other key pathways (e.g., PI3K/Akt/mTOR, Hippo/YAP), a comprehensive review should incorporate existing knowledge of their interactions with FAO into the main body. This would provide a more complete picture of the regulatory network.
- The manuscript contains an erroneous statement regarding a c-Myc feedback loop, which includes both a typographical and a citation error. The text claims, "...a positive feedback loop between EPT2 and c-Myc has been revealed... [91]". First, "EPT2" appears to be a typo and likely should be CPT2, an enzyme relevant to the paper's topic. More importantly, the cited reference [91] does not support a claim about CPT2. The paper by Liao et al. actually describes a feedback loop between c-Myc and a completely different protein, EBP2. Therefore, the statement as written is inaccurate and the citation does not support the likely intended meaning.
- There is a significant error in the citations for the drug perhexiline. The manuscript claims that perhexiline has shown efficacy in "preclinical breast and lung cancer models," citing references [107, 108]. However, these citations do not support the claim regarding lung cancer. Reference [107] is a study on chronic lymphocytic leukemia, and reference [108] investigates hepatocellular carcinoma. Neither study appears to involve lung cancer models, and this citation must be corrected.
- There is a critical error in the manuscript regarding the therapeutic agent ST1326. The text describes ST1326 as a "brain-penetrant-selective CPTIA inhibitor designed for glioblastoma treatment," with the accompanying table listing "Glioblastoma, brain metastases" as its target. However, the supporting citation [109] refers to a study titled, "A reversible carnitine palmitoyltransferase (CPT1) inhibitor offsets the proliferation of chronic lymphocytic leukemia cells". The manuscript incorrectly applies findings from leukemia research to glioblastoma, and the citation does not support the claims about either the drug's target indication or its brain-penetrant properties. This citation must be corrected.
- In Figure 1, the substrate for the System Xc- transporter is incorrectly labeled as "Cys," the abbreviation for cysteine. This transporter imports cystine, which is the oxidized dimer of cysteine. Please correct the figure to accurately label the imported molecule.
- Please adjust Figure 1 to accurately depict the beta-oxidation pathway. The diagram should clearly show that Acyl-CoA undergoes beta-oxidation to produce acetyl-CoA, NADH, and FADHâ‚‚, and that acetyl-CoA subsequently enters the TCA cycle. The way Coenzyme Q is written is incorrect. Please correct the label "CoQH2" to the standard biochemical term for its reduced form, CoQHâ‚‚. Please change the "CPT1A" label to the more general term CPT1. Since the diagram provides a general overview, it should not refer specifically to the "A" isoform.
- The legend for Figure 1 should be more descriptive. Instead of only providing a title, please expand the legend to briefly explain the key processes shown in the diagram, including the location and function of the carnitine shuttle and its relationship to the ferroptosis pathway.
- Instead of just a title, the legend should briefly explain the complex interactions shown, summarizing the relationship between the carnitine shuttle and c-Myc, as well as their roles in ferroptosis and the tumor microenvironment.
- Please add a citation in the main text to indicate which section is being summarized by Figure 1 and Figure 2.
- On line 73, the word "Morphological" is incorrectly capitalized mid-sentence. It should begin with a lowercase "m".
- It is recommended to include a simple diagram in the "Regulation of the Carnitine Shuttle" section. A schematic would make it easier for readers to clearly understand the regulatory mechanisms involved.
- The manuscript incorrectly refers to "Table 2" in the discussion of ST1326. This should be corrected to "Table 1," as this is the only table provided and it contains the relevant data.
- Please correct the abbreviation for Cancer Stem Cells in the abbreviations list. It is listed as "CSCS" but should be "CSCs" to match the correct usage throughout the text.
- On line 220, the text mentions "inhibition of GPX4c". This appears to be a typographical error and should be corrected to
- On line 516, there is a typographical error in the phrase "in additiona to". Please correct it to "in addition to".
- The reference list contains several typographical errors in author names and titles. Please carefully review and correct the entire reference section for accuracy.

Author Response
Major comments
- While the findings on the CPT1A/c-Myc feedback loop and TAM-derived L-carnitine are significant, the review's discussion in sections 4.3 and 4.4 is almost entirely dependent on a single study by Ma et al. This narrows the scope, making these sections a summary of one paper rather than a comprehensive review of the field. Integrating other related research is necessary to provide a broader and more critical perspective.
- We appreciate the reviewer’s insightful comment. I integrate previous sections 4.3 and 4.4 to new section 5.3 to explain carnitine shuttle’s influence on ferroptosis with additional explanations related to OCTN2. In new section 5.3, I add interactions of TAMs with cancer cells via JAK2/STAT3, and TAMs differentiation through ferroptosis. By the revision, I try to provide a broader perspective.
- The manuscript would be strengthened by expanding on its observation that the carnitine shuttle's role is "context-dependent." A deeper discussion is needed to explain the central paradox: why is CPT1A upregulation a survival mechanism, yet its inhibition is a viable therapeutic strategy? Explaining how a tumor's specific metabolic phenotype—such as its reliance on fatty acid oxidation (FAO) over glycolysis—creates this targetable vulnerability would offer valuable insight.
- Thank you for your valuable suggestion. I add the supporting evidence that aggressive, metastatic, and therapy-resistant cancer cells have reliance on FAO.
- The review's discussion of signaling pathways is narrowly focused on the c-Myc connection. While the "Future Research Directions" section acknowledges other key pathways (e.g., PI3K/Akt/mTOR, Hippo/YAP), a comprehensive review should incorporate existing knowledge of their interactions with FAO into the main body. This would provide a more complete picture of the regulatory network.
- Due to narrow perspective of signaling pathways related to carnitine shuttle and ferroptosis, I add new section 4.4 for interactions of signaling pathways with carnitine shuttle and ferroptosis, including Wnt/ β-catenin, Hippo/YAP, and PI3K/Akt/mTOR.
- The manuscript contains an erroneous statement regarding a c-Myc feedback loop, which includes both a typographical and a citation error. The text claims, "...a positive feedback loop between EPT2 and c-Myc has been revealed... [91]". First, "EPT2" appears to be a typo and likely should be CPT2, an enzyme relevant to the paper's topic. More importantly, the cited reference [91] does not support a claim about CPT2. The paper by Liao et al. actually describes a feedback loop between c-Myc and a completely different protein, EBP2. Therefore, the statement as written is inaccurate and the citation does not support the likely intended meaning.
- Thank you for letting me know the errors. I fix the term into EBP2.
- There is a significant error in the citations for the drug perhexiline. The manuscript claims that perhexiline has shown efficacy in "preclinical breast and lung cancer models," citing references [107, 108]. However, these citations do not support the claim regarding lung cancer. Reference [107] is a study on chronic lymphocytic leukemia, and reference [108] investigates hepatocellular carcinoma. Neither study appears to involve lung cancer models, and this citation must be corrected.
- Thank you for pointing this out. The cancer types for perhexiline are chronic lymphocytic leukemia (CLL), and hepatocellular carcinoma. I revise the cancer types in table and manuscript.
- There is a critical error in the manuscript regarding the therapeutic agent ST1326. The text describes ST1326 as a "brain-penetrant-selective CPTIA inhibitor designed for glioblastoma treatment," with the accompanying table listing "Glioblastoma, brain metastases" as its target. However, the supporting citation [109] refers to a study titled, "A reversible carnitine palmitoyltransferase (CPT1) inhibitor offsets the proliferation of chronic lymphocytic leukemia cells". The manuscript incorrectly applies findings from leukemia research to glioblastoma, and the citation does not support the claims about either the drug's target indication or its brain-penetrant properties. This citation must be corrected.
- The details of ST1326 are revised based on references, providing cancer types of Acute myeloid leukemia, and CLL.
Detailed comments
- In Figure 1, the substrate for the System Xc- transporter is incorrectly labeled as "Cys," the abbreviation for cysteine. This transporter imports cystine, which is the oxidized dimer of cysteine. Please correct the figure to accurately label the imported molecule.
- Please adjust Figure 1 to accurately depict the beta-oxidation pathway. The diagram should clearly show that Acyl-CoA undergoes beta-oxidation to produce acetyl-CoA, NADH, and FADHâ‚‚, and that acetyl-CoA subsequently enters the TCA cycle. The way Coenzyme Q is written is incorrect. Please correct the label "CoQH2" to the standard biochemical term for its reduced form, CoQHâ‚‚. Please change the "CPT1A" label to the more general term CPT1. Since the diagram provides a general overview, it should not refer specifically to the "A" isoform.
- We thank the reviewer for detail comments. Figure 1 was revised as requested, changing 'Cys' to 'Cystine,' 'CoQH2' to 'CoQHâ‚‚,' and 'CPT1A' to 'CPT1.
- The legend for Figure 1 should be more descriptive. Instead of only providing a title, please expand the legend to briefly explain the key processes shown in the diagram, including the location and function of the carnitine shuttle and its relationship to the ferroptosis pathway.
- Instead of just a title, the legend should briefly explain the complex interactions shown, summarizing the relationship between the carnitine shuttle and c-Myc, as well as their roles in ferroptosis and the tumor microenvironment.
- I add more detailed description of Figure 1 and Figure 3 (previous Figure 2) in their figure caption.
- Please add a citation in the main text to indicate which section is being summarized by Figure 1 and Figure 2.
- On line 73, the word "Morphological" is incorrectly capitalized mid-sentence. It should begin with a lowercase "m".
- I fix the incorrect capitalization.
- It is recommended to include a simple diagram in the "Regulation of the Carnitine Shuttle" section. A schematic would make it easier for readers to clearly understand the regulatory mechanisms involved.
- A schematic diagram for the regulation of the carnitine shuttle is added in section 3.2.
- The manuscript incorrectly refers to "Table 2" in the discussion of ST1326. This should be corrected to "Table 1," as this is the only table provided and it contains the relevant data.
- Please correct the abbreviation for Cancer Stem Cells in the abbreviations list. It is listed as "CSCS" but should be "CSCs" to match the correct usage throughout the text.
- On line 220, the text mentions "inhibition of GPX4c". This appears to be a typographical error and should be corrected to
- On line 516, there is a typographical error in the phrase "in additiona to". Please correct it to "in addition to".
- Thank you for detailed feedback for errors. I fix the errors of 8-11.
- The reference list contains several typographical errors in author names and titles. Please carefully review and correct the entire reference section for accuracy.
- The entire reference lists are revised.

Reviewer 2 Report
The proposed manuscript, titled “Carnitine Shuttle and Ferroptosis in Cancer”, provides a comprehensive and well-structured review of the intersection between metabolic reprogramming and regulated cell death in cancer biology. The authors explore how the carnitine shuttle system, particularly the role of carnitine palmitoyltransferase 1A (CPT1A), modulates sensitivity to ferroptosis—a form of iron-dependent lipid peroxidation-driven cell death. The review covers the molecular components and regulation of both the carnitine shuttle and ferroptosis, integrating these mechanisms within the broader context of cancer progression, cancer stem cells (CSCs), tumor-associated macrophages (TAMs), and immune evasion.
The review demonstrates an extensive grasp of the literature and effectively connects metabolic pathways to cell death sensitivity, with a special emphasis on translational relevance. Recent findings—such as the CPT1A/c-Myc positive feedback loop, the metabolic support provided by L-carnitine from TAMs, and their implications for ferroptosis resistance—are presented in detail and discussed with clinical context. The manuscript also outlines current and investigational therapeutic agents targeting the carnitine shuttle, including small-molecule inhibitors, and offers insight into combination strategies with immunotherapy. Furthermore, it includes proposed directions for future research, such as lipidomic analyses and omics-based patient stratification.
The structure of the manuscript is clear, with well-defined sections that progress logically from basic mechanisms to therapeutic applications. Figures and tables are used effectively to illustrate key concepts.
Despite these strengths, the manuscript lacks a formal description of the methods used to identify and select the literature, and there is limited critical appraisal of the strength of the evidence discussed. Additionally, the review could benefit from more explicit discussion of conflicting data, research limitations, and potential challenges to clinical translation.
Minors points to be improved
- The manuscript does not describe how the reviewed studies were selected. There is no mention of search strategies, databases used, timeframes, inclusion/exclusion criteria, or quality assessment procedures. A dedicated section explaining this would enhance transparency and reproducibility.
- The review presents the CPT1A-ferroptosis relationship as largely established but does not discuss potential variability across cancer types or the existence of contradictory findings in the literature. Including such discussion would provide a more balanced perspective.
- Some conclusions about the potential of CPT1A inhibitors or L-carnitine modulation in clinical settings may be overstated given the current preclinical status of much of the data. The authors should be more cautious when extrapolating to therapeutic efficacy and safety.
- Finaly, potential resistance mechanisms to ferroptosis-inducing therapies, such as activation of compensatory antioxidant systems or metabolic rewiring, are not discussed but should be acknowledged as a significant translational barrier.
Author Response
- The manuscript does not describe how the reviewed studies were selected. There is no mention of search strategies, databases used, timeframes, inclusion/exclusion criteria, or quality assessment procedures. A dedicated section explaining this would enhance transparency and reproducibility.
- We thank the reviewer for this valuable suggestion. The methods sections is added to explain search strategies, databases, inclusion/exclusion criteria, and quality assessment procedures. However, this review is not following systematic review process.
- The review presents the CPT1A-ferroptosis relationship as largely established but does not discuss potential variability across cancer types or the existence of contradictory findings in the literature. Including such discussion would provide a more balanced perspective.
- We appreciate the reviewer’s constructive suggestion. As research on the CPT1A-ferroptosis relationship is still limited, I've noted this limitation in the text.
- Some conclusions about the potential of CPT1A inhibitors or L-carnitine modulation in clinical settings may be overstated given the current preclinical status of much of the data. The authors should be more cautious when extrapolating to therapeutic efficacy and safety.
- Thank you for the helpful suggestion. We note the clinical challenges in section 6.1 as followings; ‘However, challenges still remain for clinical translation, linking ferroptosis and carnitine shuttle system to treat cancers. Therapeutic efficacy and safety are needed to be implicated.’
- Finaly, potential resistance mechanisms to ferroptosis-inducing therapies, such as activation of compensatory antioxidant systems or metabolic rewiring, are not discussed but should be acknowledged as a significant translational barrier.
- We thank the reviewer for the helpful suggestion. The potential resistance mechanisms are added in section 6.3.

Reviewer 3 Report
The aim of this review is a discussion underlying the molecular mechanisms between the carnitine shuttle and ferroptosis in cancer, the role of L-carnitine derived from the tumor micro-environment, and therapeutic potential of targeting the carnitine shuttle to overcome ferroptosis resistance in cancer treatment. Overall this review provides an overview of the current knowledge.
Minor:
The figures 1 and 2 are not referenced in the main review.
Line 39: typo “Morphological” instead of “morphological”.
Line 42: “are” instead of “is”
Line 160: PUFA is not defined.
Line 186: Mitochondria (is plural) serve as … instead of serves
Line 257/8: It las been shown over the last 15 years that mitochondrial permeability transition is a process happening in the inner mitochondrial membrane. Currently the ATP synthase and the adenine nucleotide translocase are the best candidates to fulfill the criteria to be the pore. The reference number 80 does not provide any information regarding the nature of the pore.
Line 293: “the” before oncogenic transcription is missing.
References: I think, the used reference program has a mind of its own! It replaced numerous special characters in the names of the referenced authors or the titles of their publications with some random other symbols. Please see reference 74 or 82 of this manuscript for some extreme examples. Please change.
Author Response
- The figures 1 and 2 are not referenced in the main review.
- Thank you for pointing this out. We revised the manuscript to reference the figures.
- Line 39: typo “Morphological” instead of “morphological”. Line 42: “are” instead of “is”. Line 160: PUFA is not defined. Line 186: Mitochondria (is plural) serve as … instead of serves. Line 293: “the” before oncogenic transcription is missing.
- Line 257/8: It las been shown over the last 15 years that mitochondrial permeability transition is a process happening in the inner mitochondrial membrane. Currently the ATP synthase and the adenine nucleotide translocase are the best candidates to fulfill the criteria to be the pore. The reference number 80 does not provide any information regarding the nature of the pore.
- We thank the reviewer for the detailed comments for 2 and 3. They are revised in the manuscript.
- References: I think, the used reference program has a mind of its own! It replaced numerous special characters in the names of the referenced authors or the titles of their publications with some random other symbols. Please see reference 74 or 82 of this manuscript for some extreme examples. Please change.
- Thank you for your suggestion. They are revised in the manuscript with reference program.

Round 2
Reviewer 1 Report
No major comment
There appears to be a discrepancy regarding the citations for perhexiline in CLL/HCC. While the text refers to references [105, 106], the table lists references [107] and [108] for perhexiline. Please confirm which set of references is accurate for perhexiline in the table.
Author Response
There appears to be a discrepancy regarding the citations for perhexiline in CLL/HCC. While the text refers to references [105, 106], the table lists references [107] and [108] for perhexiline. Please confirm which set of references is accurate for perhexiline in the table.
- Thank you for your valuable comment. I find that the references of table was not edited, and therefore, I revised the reference for the medication table. The reference list is also revised.